# Treatment, recurrence rates and follow-up of Tenosynovial Giant Cell Tumor (TGCT) of the foot and ankle—A systematic review and meta-analysis

**M. Siegel**[1]*, **L. Bode**[1], **N. Südkamp**[1], **J. Kühle**[1], **J. Zwingmann**[1,2], **H. Schmal**[1,3], **G. W. Herget**[1,4]

1 Department of Orthopedics and Trauma Surgery, Faculty of Medicine, Medical Centre–University of Freiburg, Freiburg, Germany, 2 Department of Orthopaedic Surgery and Traumatology, St. Elisabeth Hospital, Ravensburg, Germany, 3 Department of Orthopaedic Surgery, University Hospital Odense, Odense C, Denmark, 4 Comprehensive Cancer Center Freiburg CCCF, Faculty of Medicine, Medical Centre—University of Freiburg, Freiburg, Germany

* markus.siegel@uniklinik-freiburg.de

**Data Availability Statement:** All relevant data are within the paper and its Supporting Information files.

## Abstract

### Background

The tenosynovial giant cell tumor (TGCT) is a usually benign lesion which arises from the synovium. It affects joints, tendon sheaths and bursae. The clinical course is often unpredictable, and local recurrences frequently occur. The aim of this study was to describe different treatment options, surgical complications, and to develop a follow-up regime based on a systematic literature review and meta-analysis of foot and ankle lesions.

### Methods and results

1284 studies published between 01/1966 and 06/2021 were identified. 25 met the inclusion criteria, with a total of 382 patients. Of these, 212 patients had a diffuse (dTGCT) and 170 a localized (lTGCT) TGCT. Patients with a dTGCT had a mean age of 36.6±8.2 years, and 55% were female. The overall complication rate was 24% in dTGCT, irrespective of the therapeutic procedure; the mean follow-up was 37.9±27.4 months with a recurrence rate of 21%, and recurrences occurred between 3 and 144 months, the vast majority (86%) within the first 5 years following intervention. Patients with a lTGCT had a mean age of 31.2±5.7 years, and 53% were female. Complications occurred in 12%. The mean follow-up was 51.1 ±24.6 months, the recurrence rate was 7%, and recurrence occurred between 1 and 244 months after intervention.

### Conclusion

Diffuse TGCTs of the foot and ankle region have a remarkable recurrence rate irrespective of therapeutic procedures, and most lesions reoccurred within 5, with more than half of these in the first 2 years. The lTGCTs are well treatable lesions, with a low recurrence and a moderate complication rate. Based on these findings, we propose a follow-up regime for the

**Funding:** The article processing charge was funded by the Baden-Wuerttemberg Ministry of Science, Research and Art and the University of Freiburg in the funding programme Open Access Publishing. The funders had no role in study design, data collection and analysis, decision to publish, or preparation of the manuscript.

**Competing interests:** The authors have declared that no competing interests exist.

dTGCT including a clinical survey and MR imaging 3 months after surgical intervention (baseline), followed by twice-yearly intervals for the first 2 years, yearly intervals up to the fifth year, and further individual follow-up due to the fact that recurrences can even occur for years later. For the lTGCT a clinical survey and MRT is proposed after 3–6 months after intervention (baseline), followed by annual clinical examination for 3 years, and in case of symptoms MR-imaging. Larger prospective multi-center studies are necessary to confirm these results and recommendations.

## Introduction

The tenosynovial giant cell tumor TGCT, formerly known as pigmented villonodular synovitis (PVNS), is a rare, usually benign lesion of the synovium, which affects joints, tendon sheaths and bursae. It is divided according to site (intra- versus extraarticular), to the affected location, and growth pattern (localized (lTGCT), versus diffuse (dTGCT)) [1].

The incidence rates range from estimated 30.3–39 per million person-years for lTGCT and 4–8.4 per million person-years for dTGCT [2, 3], but despite the rarity of these tumors, the diffuse and the localized form are one of the most common soft-tissue tumors of the foot and ankle region [4, 5]. In both forms patients are mainly affected in their 3rd - 5th decade of life [6, 7].

Diffuse TGCTs mainly occur in the knee (64%), ankle (14%), hip (10%), feet (5%) and shoulder (1%) [8]. The lTGCT typically affects smaller joints, like digits [3, 9], and extra-articular manifestations are more common in lTGCT [10]. Localized lesions in the foot and ankle region often appear in the dorsal forefoot and in the immediate vicinity of the first interphalangeal joints [11, 12]. However, both forms can arise from any part of the foot and ankle region [6].

The lTGCT usually are small (0.5-4cm) and manifests as a solitary, lobulated and well circumscribed lesion [10]. The dTGCT are usually large, firm, or sponge-like. The villous pattern is usually present in intra-articular tumors, whereas extra-articular tumors have a multinodular appearance [10]. Histology reveals the presence of mononuclear cells, multinucleated giant cells, macrophages, inflammatory cells and hemosiderin deposition [13].

Surgical excision is still the gold standard for the treatment of the lTGCT and dTGCT, respectively [14–17]. But despite various surgical techniques, including open, arthroscopic and combined approaches with or without radiotherapy, all approaches are associated with a noticeable recurrence rate, especially in the dTGCT [14–17]. Current research also focuses on drug therapy using CSF-1R inhibitors, which seems to be a promising alternative to interventional procedures, especially in recurrent and/or unresectable lesions [18, 19].

The purpose of this systematic literature review with a meta-analysis was to assess and summarize data for foot and ankle TGCTs, including the different therapeutic treatments, complication, and recurrence rates for the diffuse and localized form and to develop a follow-up regime.

## Material and methods

### Protocol and objective

This systematic review was conducted and reported in accordance with the Preferred Reporting Items for Systematic Reviews and Meta-Analyses (PRISMA) Statement [20].

## Search methods

Two blinded authors (MS and LB) independently applied the search strategy between January 1966 and June 2021. Clinical studies were extracted from the MEDLINE, PreMEDLINE, EBM Reviews, Cochrane Database, CINAHL and EMBASE. The chosen systematic search strategy included the search terms 'pigmented villonodular synovitis', 'PVNS', 'giant cell tumor of the tendon sheat' and 'tenosynovial giant cell'. The search was additionally supplemented with component terms as well as the conjunctions "and" or "or". The authors also examined the reference lists of the identified records.

Inclusion criteria were studies in English or German language about treatments of diffuse or localized TGCT and PVNS, respectively, 2 or more subjects per study, distinction of the forms (dTGCT vs. lTGCT), and localization (foot and ankle). Exclusion criteria were studies in other languages, case reports, studies with less than 2 subjects, reviews of former studies, radiological studies, biomechanical studies, anatomical studies, and descriptive studies about surgical techniques.

## Data extraction and synthesis

In total, 1284 studies were identified. Of these, 1188 were excluded according to the in- and exclusion criteria. After the screening, 103 studies were further evaluated and 25 studies with 382 patients were found to be appropriate for analysis (**Fig 1**).

Authors' names, year of publication, level of evidence (according to the Oxford Centre for Evidence-based Medicine) [21] and number of patients were noted. In studies examining both the localized and diffuse TGCTs, patients were listed separately according to the underlying form. Epidemiological data, age, sex, localization, treatment, complications, number of recurrences and time to recurrence, were noted.

The technique of treatment was divided into open surgical, arthroscopic surgical, combined open and arthroscopic, and radiotherapy. By means of additive treatments, adjuvant radiotherapy or radiosynviorthesis (RSO), usually with Yt90, was also noted. In a study by Tsukamoto et al, the subgroups lTGCT and dTGCT were not further elaborated upon to answer their specific study question about the effect of incomplete resection on the recurrence rate. Therefore, only general data on the recurrence rate of the two types of TGCTs could be obtained and included in our analysis [22].

## Statistics

Statistical analysis was performed using Excel Version 16.43 and IBM SPSS (Version 27.0.0). A meta-analysis was done with the open statistical environment R (R Studio Version 1.2.5033) [23, 24]. Study characteristics and patient demographics were described using statistics such as weighted averages and percentages.

We used random effects meta-analysis to estimate the pooled rate of recurrences of lTGCTs and dTGCTs, the rate of recurrences for the different types of therapy within both groups, and for the complications of both entities. We created forest plots to show the results. Heterogeneity was expressed as I-squared. In addition, the prediction interval was simulated [25].

# Results

## Study characteristics and quality assessment

12 studies referred to dTGCTs [14, 15, 17, 26–34], 3 study examined subjects with lTGCTs [11, 12, 35] and 10 studies described the diffuse and localized form, respectively [9, 16, 19, 22, 36–

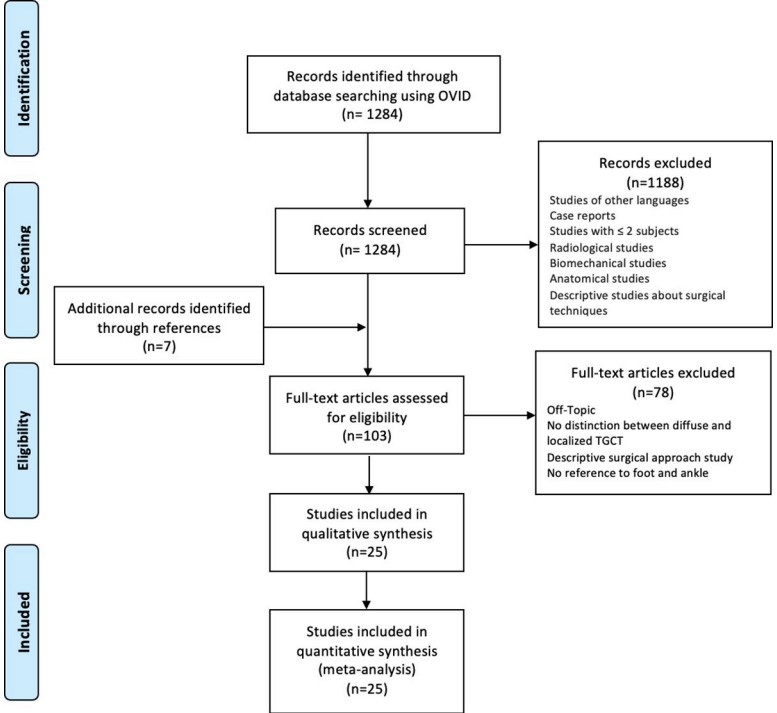

**Fig 1. Flowchart of the search strategy and study inclusion, in accordance with the PRISMA statement.**

41]. In three studies, locations besides the foot and ankle were examined. From these, the cases that met the inclusion criteria for this study were extracted [1, 21, 38].

Regarding the level of evidence, all the studies (apart from one prospective study) were retrospective, and thus reached Level IV evidence.

## Patient characteristics

Of the 382 patients, 212 had a dTGCT and 170 a lTGCT. The average number of patients per study for dTGCTs was 9.64 (range 2 to 40 subjects, and an average number of 13.1 subjects (range 3 to 44 subjects) was found for lTGCTs.

There was a mean age of 36.6 ± 8.2 years and a gender distribution of 66 females and 54 males (f:m = 1:0.8), meaning 55.0% female patients of the study population for dTGCTs. For lTGCTs, the mean age was 31.2 ± 5.7 years. 56 were female and 50 male (f:m = 1:0.9). An overview about patient characteristics per study is given in **Table 1**.

## Treatment

For patients with dTGCTs, a total of 129 subjects (13 studies) underwent open surgery [15, 16, 26, 28, 30, 32, 34, 37, 39–41], 18 patients (4 studies) underwent arthroscopic procedures [19, 26, 32, 34] and 22 patients underwent open surgery with adjuvant radiotherapy or radiosynoviorthesis [9, 14, 17, 27, 33, 36]. Two studies described patients who received arthroscopic surgery with adjuvant radiotherapy (11 patients) [27, 38], one study analyzed subjects treated with combined surgery and adjuvant radiotherapy (15 patients) [38], and 5 patients treated with radiotherapy alone (radiotherapy 15–25 fractions, 35–50 Gy) [31]. 4 Patients with dTGCTs were treated conservatively, without interventional treatment [34]. For 18 patients

**Table 1. Patient demographics and study characteristics.**

| Author | Year of publication | No. (total) | No. of dTGCT /lTGCT | mean age (y), dTGCT /lTGCT | sex, f:m, dTGCT /lTGCT, | level of evidence |
|---|---|---|---|---|---|---|
| *Spierenburg et al.* | 2021 | 84 | 40 / 44 | 38.3* | 45:39* | IV |
| *Iakovou et al.* | 2020 | 5 | 5 / 0 | 28.2 / - | 3:2 / - | IV |
| *Tsukamoto et al.* | 2020 | 33 | 18 / 15 | 42.0* | 20:13* | IV |
| *Cevik et al.* | 2019 | 26 | 11 / 15 | 40 / 26 | 7:5 / 3:12 | IV |
| *Guo et al.* | 2018 | 27 | 22 / 5 | 35.0* | 15:16* | IV |
| *Muramatsu et al.* | 2018 | 5 | 5 / 0 | 47.0 / - | - | IV |
| *Li et al.* | 2017 | 15 | 15 / 0 | 35.0 / - | 6:9 / - | IV |
| *Kanatl et al.* | 2017 | 4 | 0 / 4 | - / 27.0 | - / 2:2 | IV |
| *Cattelan et al.* | 2016 | 6 | 2 / 4 | 58.0 / 37.0 | 0:2 / 1:3 | IV |
| *Sung et al.* | 2015 | 10 | 7 / 3 | 38.4 / 36.0 | 2:5 / 1:2 | IV |
| *Korim et al.* | 2014 | 30 | 8 / 22 | 37.0* | 5:3 / 9:3 | IV |
| *Stevensons et al.* | 2013 | 18 | 18 / 0 | 42.0 / - | 11:7 / - | I |
| *Zhang et al.* | 2013 | 20 | 0 / 20 | - / 38.7 | - / 14:6 | IV |
| *Bickels et al.* | 2008 | 7 | 7 / 0 | 31.0 / - | - | IV |
| *Lee et al.* | 2006 | 7 | 7 / 0 | 30.7 / - | 3:4 / - | IV |
| *Sharma et al.* | 2006 | 14 | 5 / 9 | 28.0 / 25.5 | 2:3 / 6:3 | IV |
| *Bisbinas et al.* | 2004 | 9 | 2 / 7 | 36.0 / 22.7 | 2:0 / 7:0 | IV |
| *Saxena et al.* | 2004 | 10 | 5 / 5 | 46.0 / 35.0 | 1:4 / 3:2 | IV |
| *Brien et al.* | 2004 | 11 | 11 / 0 | 33.7 / - | 7:4 / - | IV |
| *Shabat et al.* | 2002 | 3 | 3 / 0 | 31.6 / - | 1:2 / - | IV |
| *Gibbons et al.* | 2002 | 17 | 0 / 17 | - / 29.2 | - / 10:7 | IV |
| *Ghert et al.* | 1999 | 6 | 6 / 0 | 37.3 / - | 6:0 / - | IV |
| *Rochwerger et al.* | 1999 | 8 | 8 / 0 | 47.0 / - | 4:4 / - | IV |
| *O'Sullivan et al.* | 1995 | 5 | 5 / 0 | 31.2 / - | 2:3 / - | IV |
| *Klompmaker et al.* | 1990 | 2 | 2 / 0 | 18.1 / - | 1:1 / - | IV |
| *Summary* | - | 382 | 212 / 170 | 36.6 / 31.2 | 66:54 / 56:50** | - |

no. = number, N/A = not available, dTGCT = diffuse tenosynovial giant cell tumor, lTGCT = localized tenosynovial giant cell tumor, mo = month, y = year, f = female, m = male

*for all subjects

**ratio only of specified values.

with dTGCTs, the specific therapy could not be assigned with certainty, and they were therefore excluded from the therapeutic subgroup analysis.

For patients with lTGCTs, 11 of the 13 studies described open surgical therapies for a total of 138 patients [9, 11, 12, 16, 19, 36–41]. 4 studies described arthroscopic procedures in 15 patients [19, 35, 38, 41]. In one study, an open surgical procedure with adjuvant synoviorthesis (isotopic synoviorthesis, hexatrione) was described [9]. In patients with lTGCTs, it was not possible to precisely allocate the therapy performed in a total of 15 patients.

## Complications

Complications (in particular, osteoarthritis, osteochondrosis dissecans and cartilage damage, scarring problems, draining sinus, skin necrosis, ankle stiffness, lymphedema, paraesthesia, and wound infections) were assessed in 14 studies dealing with dTGCTs and in 8 studies dealing with lTGCTs. For dTGCTs, there were 31 complications (range 0 to 7 complications per study) other than recurrences in 150 patients (70.8% of the study population for dTGCT), resulting in 24% (95% CI, 0.10 to 0.48) with a heterogeneity of I-squared = 79% (**Fig 2**).

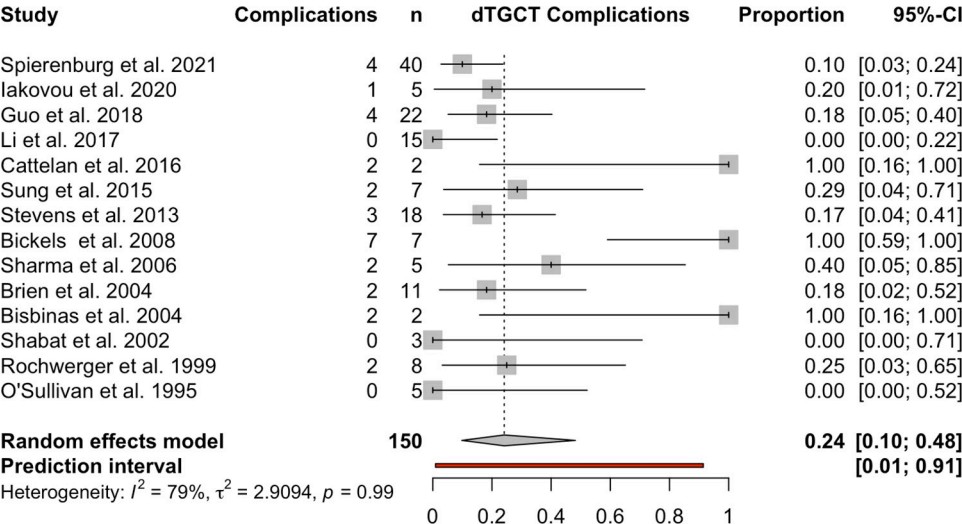

**Fig 2. Postoperative complications in patients with dTGCTs.**

The study by Bickels et al. must be critically highlighted here, who experienced unacceptable complication rates for intra-articular Yttrium 90 injections in their analysis, thus showing a 100% complication rate [14]. An overview about recurrences, the time interval to recurrences and the complications per study is given in **Table 2**.

For subjects with lTGCTs, there were 13 complications (ranging from 0 to 5 complications per study) in 107 patients (62.9% of the study population for lTGCTs) in 8 studies, resulting in a complication rate of 12% (95% CI, 0.07 to 0.20) with a 0% I-squared heterogeneity (**Fig 3**).

## Follow-up and recurrences

The mean follow-up of patients with dTGCTs ranged from 13.0 to 111.3 months, with a mean follow-up of 37.9 ± 27.4 months. For patients with lTGCTs, the mean follow-up ranged from 20.6 to 72.1 months, with a mean follow-up of 51.1 ± 24.6 months across all studies. In total, 76 recurrences were found among the 382 patients. Of these, 63/76 recurrences were found in the dTGCT group and 13/76 recurrences in lTGCTs.

## Diffuse TGCT

In the 22 studies (2 to 40 patients per study), the combined recurrence rate according to the random effects model was 21% (95% CI, 0.12–0.35), with a 67% I-squared heterogeneity (Fig 4).

To assess the recurrence rates of the different therapies in the dTGCT group, a meta-analysis of the different therapeutic subgroups was performed.

Thirteen studies focused on open surgical treatment of dTGCTs alone, with a total of 129 patients included (2 to 39 patients per study) [15, 16, 26, 28–30, 32, 34, 37, 39–41]. Among these patients, the recurrence rate according to the random effects model was 28% (95% CI, 0.17–0.44) with a 53% I-squared heterogeneity. 4 studies described 18 patients treated with arthroscopic synovectomy (1 to 11 patients per study) [19, 21, 32, 34]. Within this group, 8/18 patients had a recurrence over the course of their treatment. Due to the small number of patients, a high heterogeneity of I-squared (74%) was apparent. The statement on a recurrence rate using the random effect model (0.27% with a 95% CI, 0.02–0.88) was not appropriate.

**Table 2.  Follow-up, recurrences, time to recurrence and complications.**

| Author(s) | Year of publication | No. of Cases dTGCT/lTGCT | Mean follow-up (mo.), dTGCT / lTGCT | No. of Recurrences, dTGCT/lTGCT | Mean Time until recurrence (m) dTGCT/lTGCT | Complications dTGCT/lTGCT |
|---|---|---|---|---|---|---|
| *Spierenburg et al.* | 2021 | 40/44 | 46.5* | 23/4 | 25/84.5 | 4/5 |
| *Iakovou et al.* | 2020 | 5/0 | 47.0/- | 0/- | -/- | 1/- |
| *Tsukamoto et al.* | 2020 | 18/15 | 32.0* | 8/2 | -/- | -/- |
| *Cevik et al.* | 2019 | 11/15 | 74.5/72.1 | 3/1 | 49/24 | -/- |
| *Guo et al.* | 2018 | 22/5 | 41.0/71.0 | 5/- | 28.5/- | 4/0 |
| *Muramatsu et al.* | 2018 | 5/0 | 34.0/- | 0/- | -/- | -/- |
| *Li et al.* | 2017 | 15/0 | 37.4/- | 2/- | -/- | 0/- |
| *Kanatlia et al.* | 2017 | 0/4 | -/57.0 | -/0 | -/- | -/- |
| *Cattelan et al.* | 2016 | 2/4 | 106/64 | 1/1 | 48/108 | 2/1 |
| *Sung et al.* | 2015 | 7/3 | 38.6/20.6 | 3/1 | 7/3 | 2/0 |
| *Korim et al.* | 2014 | 8/22 | 63.7* | 2/0 | -/- | -/- |
| *Stevensonset al.* | 2013 | 18/0 | 61.0/- | 0/- | -/- | 3/- |
| *Zhang et al.* | 2013 | 0/20 | 3–60** | -/4 | -/5.3 | -/1 |
| *Bickels et al.* | 2008 | 7/0 | - | 0/- | -/- | 7/- |
| *Lee et al.* | 2006 | 7/0 | 24.0/- | 0/- | -/- | -/- |
| *Sharma et al.* | 2006 | 5/9 | 31.5 / 63.4 | 2/0 | 35.1/- | 2/3 |
| *Bisbinas et al.* | 2004 | 2/7 | 15.0 / 34.0 | 2/0 | 55.1/- | 2/- |
| *Saxena et al.* | 2004 | 5/5 | 46.0 / 57.6 | 2/0 | -/- | -/0 |
| *Brien et al.* | 2004 | 11/0 | 111.3/- | 7/- | 18/- | 2/- |
| *Shabat et al.* | 2002 | 3/0 | 64.0/- | 0/- | -/- | 0/- |
| *Gibbons et al.* | 2002 | 0/17 | -/85 | -/0 | -/- | -/3 |
| *Ghert et al.* | 1999 | 6/0 | 13.0/- | 1/- | 48/- | -/- |
| *Rochwerger et al.* | 1999 | 8/0 | 48.4/- | 1/- | 96/- | 2/- |
| *O'Sullivan et al.* | 1995 | 5/0 | 34.8/- | 0/- | -/- | 0/- |
| *Klompmaker et al.* | 1990 | 2/0 | 39.0/- | 1/- | 5/- | -/- |
| ***Summary*** | - | **382** | **37.9 / 51.1** | **63/13** | **28.3 / 44.9** | **31/13** |

No = number; m = month(s); y = year(s)

* for all subjects

** a mean was not provided.

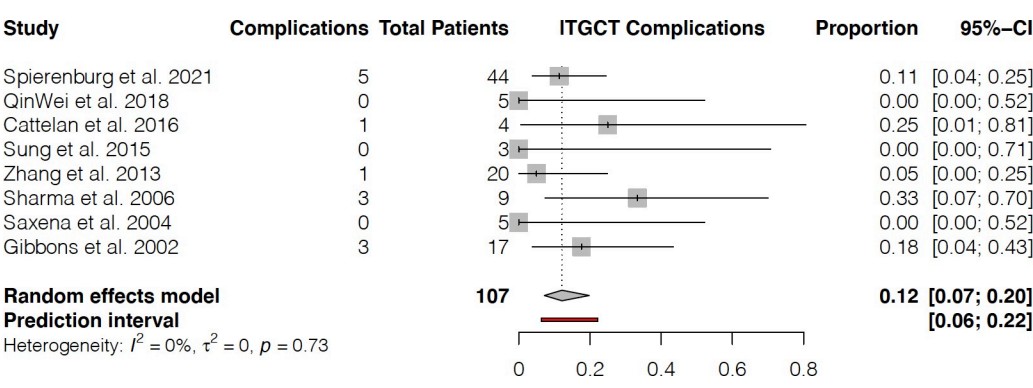

**Fig 3. Complications in patients with lTGCTs.**

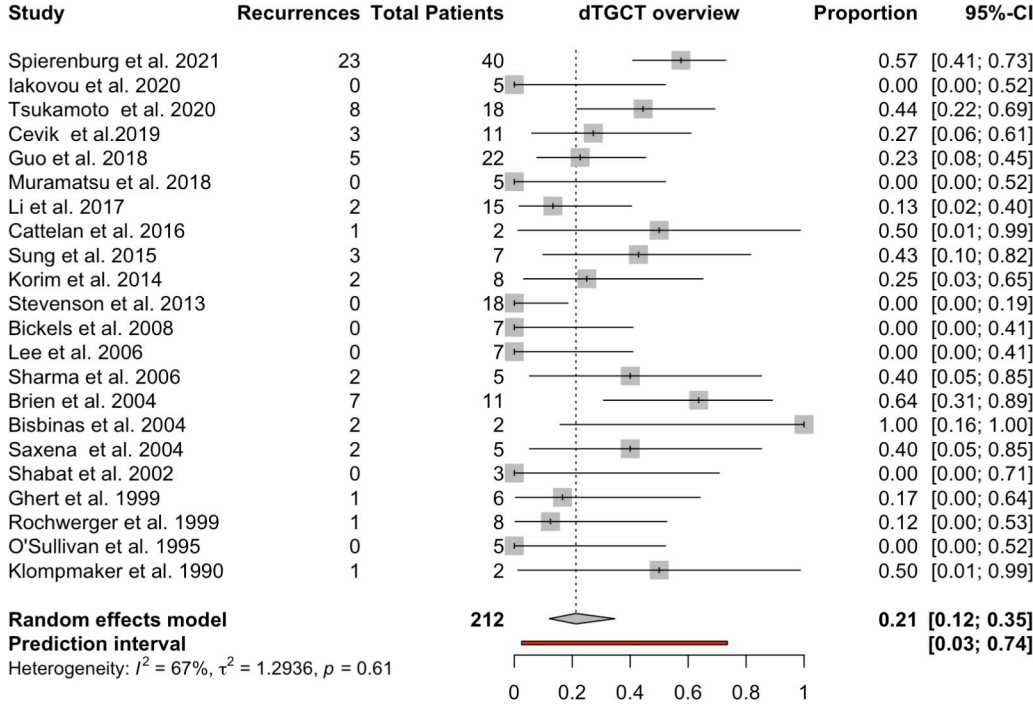

**Fig 4. Recurrences in patients with dTGCTs.**

22 Subjects with open surgery and radiotherapy were found in 6 studies, with 2 to 7 patients per study [9, 14, 17, 27, 33, 36].

Within this subgroup, the combined recurrence rate according to the random effect model was 4% (95% CI, 0.04–0.04). Heterogeneity was also high in this subgroup, with I-squared at 88%.

Further meta-analysis of the therapeutic subgroups (i.e. combined arthroscopy with radiotherapy, combined open and arthroscopic surgery with or without radiotherapy, sole radiotherapy) in the patients with dTGCTs was not carried out, due to a lack of patient numbers and therefore a lack of statistical significance.

## Localized TGCTs

In nine studies (3 to 44 patients per study), 13 recurrences occurred in 170 patients [9, 11, 12, 16, 19, 36–41]. This resulted in a combined recurrence rate according to the random effect model of 7% (95% CI, 0.03–0.15), irrespective of the therapeutic approach. The heterogeneity resulted in an I-squared value of 17% (**Fig 5**). A further investigation of the therapeutic subgroups was also not carried out here, due to the low overall number of recurrences.

## Time to recurrences

For the determination of the time to recurrence for patients with dTGCTs, the data for 50/63 patients with recurrences could be included (79.3%), for 13 patients the exact time to recurrence was not given. Patients with dTGCTs had a mean time to recurrence of 33.8 months (ranging from 3 to 144), irrespective of the therapeutic approach. Recurrences were more frequent in the first two years, and a second peak was found around 4 years after therapy. Within the first 12 months after the therapeutic intervention, 28% of all recurrences occurred and

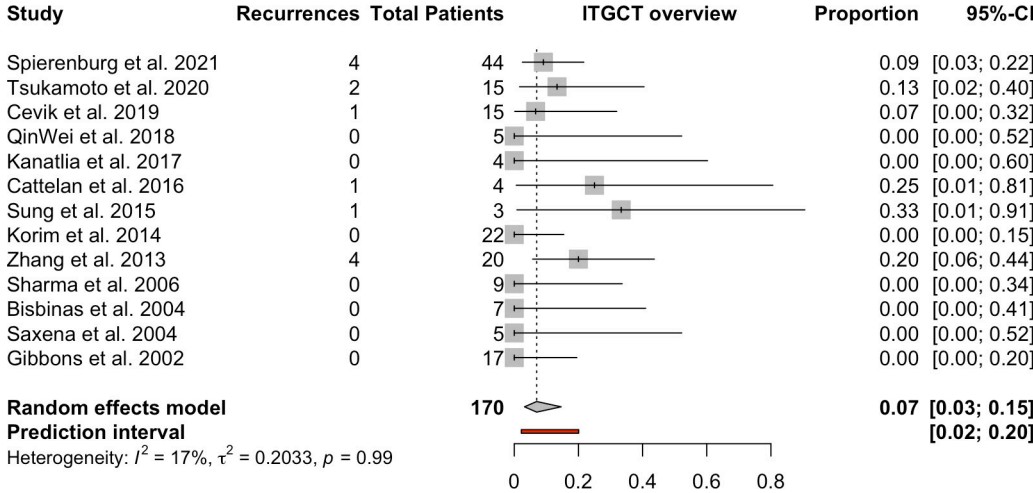

**Fig 5. Recurrences in patients with lTGCTs.**

another 24% could be diagnosed 24 months post-intervention, in summary more than half of all recurrences. Between 25 and 36 months after the intervention there were only four additional recurrences. This was before a further accumulation of recurrences occurred in the fourth year, with a total of 9 representing another 18% of all recurrences. Three recurrences were described within the 5th and two recurrences in the 6th years after intervention. Five late recurrences were found, after more than 84 months, with the last recurrence after 144 months (**Fig 6**).

Patients with lTGCTs showed a low recurrence rate (7%), with a total of 13 recurrences in 170 cases. Time to recurrence was provided in 11/13 cases (84.6%). In this group, the average time to recurrence was 55.6 ± 83.6 months, ranging from 1 to 264 months. Within the first 3 years, 8 of the 11 recurrences occurred (72.7%). In 3 cases, the recurrences occurred after 9 and 12 years, and with a very late event after 22 years (**Fig 6**).

## Discussion

This systematic literature review with a meta-analysis showed a remarkable recurrence rate for patients with dTGCTs (21%) and a high complication rate (24%), regardless of the therapeutic

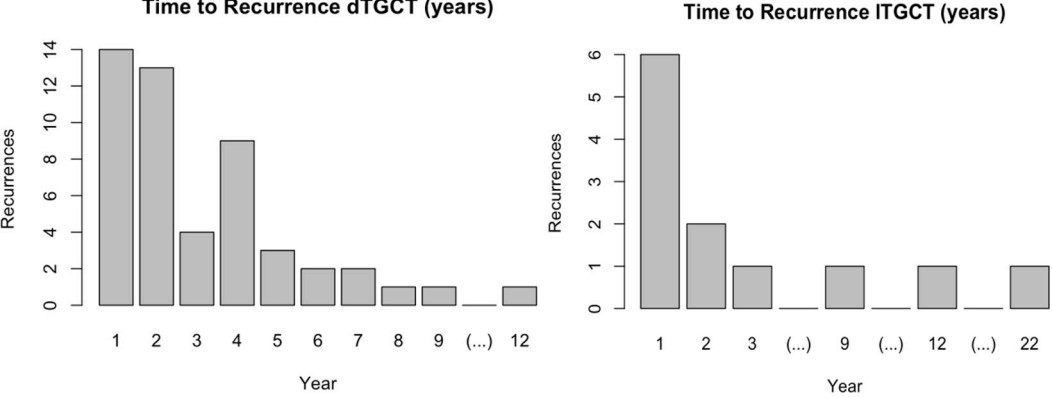

**Fig 6. Recurrences (n = 50/63) of diffuse tenosynovial giant cell tumors (dTGCTs), and recurrences (n = 11/13) of localized tenosynovial giant cell tumors (lTGCTs).**

approach. In patients with lTGCTs, the recurrence rate was 7% and a moderate complication rate of 12% was found, independent of the conducted therapy.

Unlike with existing data for TGCTs of the knee or hip, there are only a few studies that refer to foot and ankle TGCTs, probably due to the low frequency incidence [42]. Our systematic review, which to the best of our knowledge includes the largest number of subjects with TGCTs in the foot and ankle, confirms the suggested slight predisposition for females in other affected locations, with 55% of female patients with dTGCTs and 53% with lTGCTs in the foot and ankle [8, 42, 43].

The age distribution for patients in our collective, with a mean age of 36.6±8.2 years in patients with dTGCTs and a mean age of 31.2±5.7 years for patients with lTGCTs, also corresponds to the findings in other affected joints, with a peak incidence between 20 and 40 years of age [6, 8, 44]. Our results therefore show consistent data for the demographic characteristics of patients with TGCTs that are currently known about.

## Treatment

At present, the choice of treatment for TGCTs in the foot and ankle depends on the type of TGCT, the skill of the surgeon, and the surgical or therapeutical availability of the oncological centre. Recently, in the knee joint there has been a trend towards arthroscopic resection of lTGCTs, whereas in cases of extensive involvement especially in the case of extra-articular involvement (predominantly dTGCTs), open surgery is needed for a complete tumor resection [7, 43, 45]. For the foot and ankle region, arthroscopic resection is rather rare and sometimes technically difficult and depending on the localization and pattern of spread not promising.

In 2017, Fraser et al. conducted a review of TGCT in the foot and ankle region and attempted to recommend possible treatment options based on levels of evidence. This generated a Grade C recommendation only for open synovectomy/excision, which is supported by poor-quality evidence (level IV-V studies). For all other treatment options (arthroscopic synovectomy/excision, external beam radiation therapy, intra-articular radioactive isotope injection and targeted pharmacological treatments), there was insufficient evidence to make any treatment recommendation [6].

Recently, in the largest case series published to date on TGCT in the foot and ankle region, a recurrence rate in arthroscopically treated cases (n = 17) of 47% for lTGCT and 64% for dTGCT was reported [19]. For open resection (n = 76) a recurrence rate of 36% for dTGCT and 11% for lTGCT, respectively, was described. This resulted in the recommendation for primary open resection, especially in dTGCT with extraarticular involvement [19].

Our results show that arthroscopic treatment is rather rare, with 44 cases in the whole study population. In both dTGCTs and lTGCTs, the majority of patients were treated with open surgery.

A review by Noailles et al., published in 2017, found that arthroscopic resection in patients with lTGCTs showed high efficacy for all 4 major joints (shoulder, hip, knee and ankle). For this assessment, 6 studies were included for the evaluation of TGCTs in the ankle. In patients with dTGCTs, high effectiveness was only shown for knee joint involvement [7]. In their meta-analysis, however, Mollon et al. showed a significantly higher recurrence rate for arthroscopic dTGCT procedures in the knee joint [43].

In a recently published review, Healey et al. questioned the best possible management of patients with TGCTs and proposed a multidisciplinary therapeutic approach, including systemic therapy, due to the relatively high recurrence rates for surgical therapy approaches [45]. Due to the considerable recurrence rate of TGCTs and the remarkably high complication rate after treatment of TGCTs, especially in the foot and ankle, the choice of best therapy remains a

challenge. For multiple recurrences, systemic therapy has become a recent focus of research. This is certainly a therapeutic option to be further investigated, especially in cases of therapy-refractory disease [18].

## Complications

Complications occurred in 24% of patients with dTGCTs and in 12% of patients with lTGCTs. Based on a large collective from a study by Mastboom et al., who describe a complication rate of 12% after surgery in patients with dTGCTs regardless of the affected body region, a higher potential for complications can be assumed for involvement of the foot and ankle [8].

Mollon et al., who subdivided the various complications in the knee joint region into subgroups, showed a lower complication rate for TGCTs of the knee (stiffness 6.8%, wound infection 0.8%) than our results [43].

One explanation may be the poorer blood circulation in the foot and ankle joints when compared to other joints, which can be significantly reduced by factors such as nicotine abuse, artherosclerosis and diabetes, thus reducing the healing potential [46]. The hygienic aspects of foot surgery can also sometimes be challenging. However, these factors were not included in our data collection and have therefore not been included in our analysis.

## Recurrences

Irrespective of the chosen therapy, our study results revealed a recurrence rate of 21% for patients suffering from the diffuse form and 7% for those with a localized TGCT.

Compared to the results of the meta-analysis by Mollon et al., who investigated the recurrence rates of TGCTs of the knee (27.7% for dTGCTs and 7.1% for lTGCTs), our study showed a lower recurrence rate for both the diffuse (21%) and a comparable rate for localized (7%) forms [43].

In a recently published retrospective multi-center study from 2019, Mastboom et al. assessed, among other variables, the recurrences in a total of 966 patients with dTGCTs in various joints. The total number of recurrences was 425 for all surgically treated cases. For the recurrence rate of patients with dTGCTs of the ankle and foot, the recurrence-free-survival rate at 5 years was 72% and therefore higher than subjects with dTGCTs in the knee (61%), hip (65%) and upper extremities (59%) [8].

These surveys, when considering our results, suggest a lower overall recurrence rate, at least for dTGCTs, in the foot and ankle than in the knee and other large joints. As with the knee joint, the recurrence rate of lTGCTs is substantially lower than in dTGCTs.

With regard to assist decision-making for therapy in patients with diffuse or localized TGCTs in the foot and ankle, we also wanted to collect and compare the recurrence rate for different therapeutic approaches.

For further meta-analytical analysis of the recurrence rates of different therapeutic approaches, as well as for surgical techniques with or without adjuvant radiotherapy, we found a very high heterogeneity in our collective. A breakdown into therapeutic subgroups only showed poor analytical relevance for the openly surgically treated subjects of the dTGCT group, with a slightly higher recurrence rate of 28% than the overall rate of 21%. In our opinion, further research with larger populations is needed to generate meaningful results.

## Proposal of a follow-up regime

A question that arises frequently among surgeons and oncologists is for how long and often a patient with a diffuse or localized TGCT in the foot or ankle should be followed-up on. For this, the two types of TGCTs are considered separately.

The analysis of the time-to-recurrence of subjects with dTGCTs in this study showed a clustering of recurrences in the first two years and then in the fourth year after intervention.

However, recurrences also occurred sporadically further on, with late recurrences even after more than ten years after intervention. The assumption that further recurrences are unlikely to occur after a 5-year recurrence-free interval, as with other diseases, does not apply to dTGCTs. Mastboom et al. also reported that with longer follow-up times, recurrences still continue to increase [8].

The studies included in our analysis showed inconsistencies in the chosen follow-up regime. Consensus exists, however, regarding the necessity of MRI diagnostics paired with clinical examination. Only few studies have outlined a specified schedule of clinical and radiological follow-ups, with a submitted concept for these.

In the studies included in this analysis focusing on dTGCTs, the first time point for clinical radiological follow-up ranged from 3 to 24 months. Both Korim et al. and Tsukamoto et al. defined an overall follow-up of 5 years [16, 22], with initial higher-frequency follow-ups and then annual check-ups. Other studies described a follow-up regime with mainly diagnostic imaging based on clinical symptoms in the post-interventional course [28, 36, 41]. One study described a follow-up based solely on clinical symptoms [32].

For lTGCTs, clinical-radiological controls were, most commonly, used initially and then later purely on the basis of clinical symptoms [16, 36].

According to our results and the findings in the current literature for patients with dTGCTs we recommend an initial clinical and radiological (MRI) follow-up 3 months after intervention as baseline. This would assess the success of the therapy and provide a reference point for further follow-ups. We then recommend twice-yearly clinical and radiological follow-ups for two years, followed by annual clinical and radiological examinations until the 5th year. This would detect the majority of recurrences. From the 5th year onwards, we recommend an individual follow-up based on the individual post-interventional course of the subject. Should the clinical symptoms rapidly increase after treatment, we would recommend follow-up examinations outside of this concept, according to the patient's complaints.

For patients with lTGCTs, an early clinical and radiological follow-up 3–6 months after surgical intervention is recommended to assess the success of the therapy. Despite the low recurrence rate, an annual clinical and in case of symptoms radiological (MRI) follow-up for 3 years is recommended, and subsequent, individual follow-up on the basis of the clinical symptoms.

## Limitations

This study has several limitations. Due to the rarity of dTGCTs and lTGCTs in the foot and ankle region compared to other joints, there was a comparatively small number of studies that could be included in our analysis. Some of the included studies included showed pooled recurrence rates without a division into therapeutic subgroups and could therefore not be included in a further corresponding analysis.

Further limitations of our review are due to the heterogeneity and divergent quality of the different case series included. Almost all studies were Level IV evidence, and thus low quality.

This was also evident in the complication rate, gender ratio, age, and other demographic data. Factors that could have had an influence on the recurrence and complication rate, such as comorbidities, spread patterns, the localization of the TGCT Lesion and the surgeon's skills, could not be collected homogeneously and thus could not be evaluated.

Only a few studies reported the time-to-recurrence. Consequently, our recommendation on the follow-up regime had to be based on the relatively small quantity of data available on TGCT of the foot and ankle and was also guided by subjective experience from clinical practice

and existing literature on TGCTs. It should also be noted that in the present data there is a mean follow-up time of 37.9 months for the dTGCTs and of 51.1 months for the lTGCTs with a mean time to recurrence of 28.3 months for dTGCT and correspondingly 44.9 months for lTGCT. Therefore, this calculation will always be an underestimation of the actual time to recurrence. As described by Mastboom et al., recurrences can occur far beyond the proposed follow-up scheme, which is why a critical individual follow-up must be considered, especially in high-risk patients [8].

However, with this study we can provide new data on the understanding, assessment, and treatment of TGCTs of the foot and ankle, using the largest patient sample to the best of our knowledge.

## Conclusion

Diffuse TGCTs of the foot and ankle were associated with a remarkable complication and recurrence rate irrespective of the method of surgical treatment. Most recurrences occurred up to 5 years after initial treatment, with more than half of these within 2 years, requiring a consistent follow-up for at least this period, followed by further follow-up based on the risk profile and the clinical symptoms. Localized TGCTs showed a low recurrence rate and a moderate complication rate. An annual clinical and in case of symptoms radiological follow-up for 3 years is recommended, followed by an individual follow-up on the basis of the clinical symptoms. Further validation utilizing prospectively collected data are warranted to evaluate the most promising treatment and follow-up for foot and ankle TGCTs.

## Supporting information

**S1 Checklist. PRISMA 2020 checklist.**
(DOCX)

**S1 Fig. Recurrences in patients with dTGCTs treated by open surgery.**
(TIF)

**S2 Fig. Recurrences in patients with dTGCTs treated by arthroscopic surgery.**
(TIF)

**S3 Fig. Recurrences in patients with dTGCTs treated by open surgery and radiotherapy.**
(TIF)

**S4 Fig. Recurrences in patients with lTGCTs treated by open surgery.**
(TIF)

**S5 Fig. Recurrences in patients with lTGCTs treated by arthroscopic surgery.**
(TIF)

## Acknowledgments

We would like to express our special thanks to Dr. Spierenburg and Prof. van de Sande for providing us their data regarding the time to recurrence.

## Author Contributions

**Conceptualization:** M. Siegel, L. Bode, N. Südkamp, J. Zwingmann, G. W. Herget.

**Data curation:** M. Siegel, L. Bode.

**Formal analysis:** M. Siegel.

**Funding acquisition:** M. Siegel.

**Investigation:** M. Siegel, G. W. Herget.

**Methodology:** M. Siegel, L. Bode, J. Zwingmann, G. W. Herget.

**Project administration:** M. Siegel, J. Zwingmann, G. W. Herget.

**Software:** M. Siegel.

**Supervision:** J. Zwingmann, H. Schmal, G. W. Herget.

**Visualization:** M. Siegel.

**Writing – original draft:** M. Siegel.

**Writing – review & editing:** M. Siegel, J. Kühle, J. Zwingmann, H. Schmal, G. W. Herget.

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
