## [Decision Letter · Decision Letter 0]

2 Nov 2021

PONE-D-21-30856Treatment, Recurrence Rates and Follow-up of Tenosynovial Giant Cell Tumor (TGCT) of the Foot and Ankle – A Systematic Review and Meta-AnalysisPLOS ONE

Dear Dr. Siegel,

Thank you for submitting your manuscript to PLOS ONE. After careful consideration, we feel that it has merit but does not fully meet PLOS ONE’s publication criteria as it currently stands. Therefore, we invite you to submit a revised version of the manuscript that addresses the points raised during the review process.

Specifically, please add references and correct the language errors suggested by reviewers.

We look forward to receiving your revised manuscript.

Kind regards,

Akihiko Takeuchi, M.D., Ph.D

Academic Editor

PLOS ONE

Journal Requirements:

1. Thank you for stating the following financial disclosure: 

"The author(s) received no specific funding for this work.

The article processing charge was funded by the Baden-Wuerttemberg Ministry of Science, Research and Art and the University of Freiburg in the funding programme Open Access Publishing."

e) Please provide an amended Funding Statement that declares *all* the funding or sources of support received during this specific study (whether external or internal to your organization) as detailed online in our guide for authors at http://journals.plos.org/plosone/s/submit-now.  

f) Please state what role the funders took in the study.  If any authors received a salary from any of your funders, please state which authors and which funder. If the funders had no role, please state: "The funders had no role in study design, data collection and analysis, decision to publish, or preparation of the manuscript." 

Please send your amended statements by return email; we will change the online submission form on your behalf.

Additional Editor Comments (if provided):

Reviewers' comments:

Reviewer's Responses to Questions

**Comments to the Author**

1. Is the manuscript technically sound, and do the data support the conclusions?

Reviewer #1: Yes

Reviewer #2: Yes

2. Has the statistical analysis been performed appropriately and rigorously? 

Reviewer #1: Yes

Reviewer #2: Yes

3. Have the authors made all data underlying the findings in their manuscript fully available?

Reviewer #1: Yes

Reviewer #2: Yes

4. Is the manuscript presented in an intelligible fashion and written in standard English?

Reviewer #1: Yes

Reviewer #2: Yes

5. Review Comments to the Author

Reviewer #1: I read the revised manuscript with great interest again.

I believe the authors addressed the feedback they have received before adequately.

With the new search they included more relevant papers.

Also, the claims they made are now better substantiated.

If the authors would add references to study regarding the results in lines 263 - 288, that would improve their manuscript.

Reviewer #2: The Systematic Review and Meta-Analysis is much better than previous version. I saw that you put a lot of effort into revising this systematic review.

Line 95 "... shoulder (1%) [25], The lTGCT ..." The comma should be dot.

Congratulations for this review.

6. PLOS authors have the option to publish the peer review history of their article (what does this mean?). If published, this will include your full peer review and any attached files.

Reviewer #1: No

Reviewer #2: No

---

## [Author Response · Author response to Decision Letter 0]

4 Nov 2021

Editor's notes have been edited and added, both in the Reponse-to-Reviewer letter and in the Manuscript. Files are uploaded to PACE.

The additions of References from Reviewer 1 have been completed. The punctuation, which was pointed out by Reviewer 2, has been corrected. 

Thanks a lot, 

Markus Siegel

---

## [Decision Letter · Decision Letter 1]

17 Nov 2021

Treatment, Recurrence Rates and Follow-up of Tenosynovial Giant Cell Tumor (TGCT) of the Foot and Ankle – A Systematic Review and Meta-Analysis

PONE-D-21-30856R1

Dear Dr. Siegel,

We’re pleased to inform you that your manuscript has been judged scientifically suitable for publication and will be formally accepted for publication once it meets all outstanding technical requirements.

Kind regards,

Akihiko Takeuchi, M.D., Ph.D

Academic Editor

PLOS ONE

Additional Editor Comments (optional):

Reviewers' comments:

Reviewer's Responses to Questions

**Comments to the Author**

1. If the authors have adequately addressed your comments raised in a previous round of review and you feel that this manuscript is now acceptable for publication, you may indicate that here to bypass the “Comments to the Author” section, enter your conflict of interest statement in the “Confidential to Editor” section, and submit your "Accept" recommendation.

Reviewer #1: All comments have been addressed

Reviewer #2: All comments have been addressed

2. Is the manuscript technically sound, and do the data support the conclusions?

Reviewer #1: Yes

Reviewer #2: Yes

3. Has the statistical analysis been performed appropriately and rigorously? 

Reviewer #1: Yes

Reviewer #2: Yes

4. Have the authors made all data underlying the findings in their manuscript fully available?

Reviewer #1: Yes

Reviewer #2: Yes

5. Is the manuscript presented in an intelligible fashion and written in standard English?

Reviewer #1: Yes

Reviewer #2: Yes

6. Review Comments to the Author

Reviewer #1: (No Response)

Reviewer #2: I read the revised manuscript with interest. I believe the authors addressed all the feedbacks they have received before adequately. Thank you, good job

7. PLOS authors have the option to publish the peer review history of their article (what does this mean?). If published, this will include your full peer review and any attached files.

Reviewer #1: No

Reviewer #2: No

---

## [Editor Report · Acceptance letter]

19 Nov 2021

PONE-D-21-30856R1 

Treatment, Recurrence Rates and Follow-up of Tenosynovial Giant Cell Tumor (TGCT) of the Foot and Ankle – A Systematic Review and Meta-Analysis 

Dear Dr. Siegel:

I'm pleased to inform you that your manuscript has been deemed suitable for publication in PLOS ONE. Congratulations! Your manuscript is now with our production department. 

Kind regards, 

on behalf of

Dr. Akihiko Takeuchi 

Academic Editor

PLOS ONE